# ENHANCING PROTOTYPE-BASED FEDERATED LEARNING WITH STRUCTURED SPARSE PROTOTYPES

## ABSTRACT

Prototype-Based Federated Learning (PBFL) has gained attention for its communication efficiency, privacy preservation, and personalization capabilities in resource-constrained environments. Despite these advantages, PBFL methods face challenges, including high communication costs for high-dimensional prototypes and numerous classes, privacy concerns during aggregation, and uniform knowledge distillation in heterogeneous data settings. To address these issues, we introduce three novel methods, each targeting a specific PBFL stage: 1) Class-wise Prototype Sparsification (CPS) reduces communication costs by creating structured sparse prototypes, where each prototype utilizes only a subset of representation layer dimensions. 2) Privacy-Preserving Prototype Aggregation (PPA) enhances privacy by eliminating the transmission of client class distribution information when aggregating local prototypes. 3) Class-Proportional Knowledge Distillation (CPKD) improves personalization by modulating the distillation strength for each class based on clients' local data distributions. We integrate these three methods into two foundational PBFL approaches and conduct experimental evaluations. The results demonstrate that this integration achieves up to 10× and 4× reductions in communication costs while outperforming the original and most communication-efficient approaches evaluated, respectively.

## 1 INTRODUCTION

Federated Learning (FL) has emerged as an innovative paradigm in distributed machine learning, enabling collaborative model training across decentralized devices while preserving data privacy (McMahan et al., 2017). However, FL faces significant challenges, primarily due to heterogeneous data distributions (Zhao et al., 2018; Li et al., 2022) and diverse model architectures across clients (Li & Wang, 2019; Lin et al., 2020), which often lead to performance degradation. Researchers have proposed various personalized and heterogeneous federated learning approaches to address these challenges. For data heterogeneity, approaches include model interpolation (Li et al., 2021; Deng et al., 2020; Lee & Choi, 2024), clustering (Sattler et al., 2020; Ghosh et al., 2020; Briggs et al., 2020; Duan et al., 2021), and multi-task learning (Mills et al., 2021; Hanzely & Richtárik, 2020; Huang et al., 2021). To tackle model heterogeneity, researchers have developed strategies such as logit or representation exchange on public datasets (Li & Wang, 2019; Lin et al., 2020; Zhang et al., 2021c) and partial model (Liang et al., 2020; Zhu et al., 2021) or auxiliary model sharing (Wu et al., 2022; Zhang et al., 2022).

Despite the progress made by these approaches, many existing methods are not communication-efficient, as they involve sharing large amounts of model parameters or logits on a public dataset. This makes them unsuitable for resource-constrained devices, especially those with limited bandwidth. In response to these limitations, Prototype-Based Federated Learning (PBFL) has emerged as a promising alternative (Tan et al., 2022a). PBFL significantly reduces communication overhead by transmitting only prototypes between the server and clients, with the communicated data size limited to the prototype dimension multiplied by the number of classes. Moreover, PBFL enhances privacy protection by design because prototypes represent averages of local models' representations. Furthermore, PBFL naturally facilitates personalization by allowing local models to distill knowledge exclusively from global prototypes corresponding to classes in their local datasets.

Despite these advantages, PBFL still needs to overcome several challenges that limit its effectiveness in specific scenarios. While generally more communication-efficient than other approaches, PBFL can still incur high communication costs when the dimension of the prototype is very high or the number of classes is vast. Additionally, some existing PBFL methods, such as (Tan et al., 2022a), often require the server to know each client's class distribution when aggregating local prototypes, potentially compromising privacy (Zhang et al., 2024). Another challenge is that the uniform knowledge distillation of global prototypes without considering data heterogeneity can hinder effective personalization, potentially leading to suboptimal performance.

To address these challenges and fully realize PBFL's potential in resource-constrained environments, we propose three novel methods that can be applied to existing PBFL frameworks. Class-wise Prototype Sparsification (CPS) enforces structured sparse prototypes per class by assigning specific representation dimensions to each prototype, zeroing out others. By transmitting only non-zero dimensions, CPS significantly reduces communication costs. Privacy-Preserving Prototype Aggregation (PPA) performs weighted averaging of local prototypes without requiring the server to know clients' class distributions, thereby enhancing privacy. Finally, Class-Proportional Knowledge Distillation (CPKD) distills knowledge from global prototypes by weighting the distillation process based on local class distributions. This approach facilitates effective adaptation to each client's unique data characteristics, thus improving personalization.

Our three methods have been evaluated using heterogeneous lightweight models. Experimental results demonstrate that when applied to two established PBFL approaches (FedProto and FedTGP), our methods significantly reduce communication costs while outperforming the original and several data-free FL approaches.

## 2 RELATED WORK

### 2.1 HETEROGENEOUS FEDERATED LEARNING

Heterogeneous Federated Learning (HtFL) has emerged as a response to the challenge of heterogeneity in real-world federated settings. HtFL strategies can be broadly classified into two categories: those dependent on public data and those that operate without such reliance. Public data-dependent approaches leverage shared or globally accessible datasets to facilitate knowledge transfer across heterogeneous clients. Knowledge Distillation (KD) based methods are notable examples in this category (Li & Wang, 2019; Zhang et al., 2021b; Yu et al., 2022). Data-free approaches can be categorized based on what is shared among the server and clients: partial model parameters, auxiliary model parameters, or prototypes. Partial model sharing strategies, such as LG-FedAvg (Liang et al., 2020) and FedGen (Zhu et al., 2021), partition client model architectures. By sharing only upper layers while allowing lower layers to vary, these approaches aim to balance model customization with knowledge sharing. Alternatively, auxiliary model-based techniques like FML (Shen et al., 2020) and FedKD (Wu et al., 2022) train and share a compact auxiliary model through mutual distillation. An auspicious direction in data-free HtFL is the use of prototype-based methods that share condensed class representations (Jeong et al., 2018; Tan et al., 2022a;b; Huang et al., 2023; Zhang et al., 2024). By focusing on essential class-level information, prototype-based methods aim to strike a delicate balance between effective knowledge sharing and privacy preservation.

### 2.2 PROTOTYPE-BASED FEDERATED LEARNING

(Jeong et al., 2018) employs class-wise averaged logits for knowledge transfer, which can pose privacy risks by exposing the number of classes and each class's logit distribution. To address this, FedProto (Tan et al., 2022a) introduced a more privacy-safe approach by exchanging local prototypes of the decision layer instead of logits. Building upon these foundations, several works have further refined PBFL techniques. FedTGP (Zhang et al., 2024) enhances performance through Adaptive-margin-enhanced Contrastive Learning (ACL), which refines global prototypes. To improve efficiency, FedPCL (Tan et al., 2022b) leverages both class prototypes and pre-trained models, effectively reducing computational and communication costs. Addressing the challenge of domain shift in federated learning, Federated Prototypes Learning (FPL) (Huang et al., 2023) develops cluster and unbiased prototypes, offering rich domain insights and a balanced convergence objective.

Our work builds upon these foundations, introducing novel techniques to enhance PBFL's capabilities in addressing these challenges.

## 3 PROBLEM FORMULATION

We consider a system comprising $M$ clients and a server. The clients interact with the server to jointly develop personalized models without sharing their private data directly. Each client $i$ in this HPFL setup has its data distribution $P_i$ with $K$ classes. These distributions can differ between clients, reflecting the typical scenario in HPFL. We define a loss function $\ell$ that evaluates the performance of each client's local model $\boldsymbol{w}_i$ on data points from their respective distributions. The aim of HPFL can be described as minimizing the mean expected loss across all clients:

$$\min_{\mathbf{W}} \left\{ F(\mathbf{W}) := \frac{1}{M} \sum_{i=1}^{M} \mathbb{E}_{(x,y) \sim P_i} \left[ \ell(\boldsymbol{w}_i; x, y) \right] \right\}, \tag{1}$$

where $\mathbf{W} = [\boldsymbol{w}_1, \boldsymbol{w}_2, ..., \boldsymbol{w}_M]$ represents a matrix containing all individual client models. Given that we only have a limited set of data points, we estimate this expected loss using the empirical risk calculated on each client's local training dataset $\mathcal{D}_i = (x_i^{(l)}, y_i^{(l)})_{l=1}^{n_i}$, with its corresponding empirical distribution $\hat{P}_i$. Thus, the training objective becomes finding the optimal set of local models that minimizes the average empirical risk across all clients:

$$\mathbf{W}^* = \arg\min_{\mathbf{W}} \frac{1}{M} \sum_{i=1}^{M} \mathcal{L}_i(\boldsymbol{w}_i) \tag{2}$$

Here, $\mathcal{L}_i(\boldsymbol{w}_i) = \frac{1}{n_i} \sum_{l=1}^{n_i} \ell(\boldsymbol{w}_i; x_i^{(l)}, y_i^{(l)})$ represents the average loss for each client, calculated over their private training data.

In this work, we split the deep neural network $\boldsymbol{w}_i$ of client $i$ into two parts: the representation layers (feature extractor) and the decision layer (classifier). The $i$-th client's feature extractor, denoted as $f_i$ and governed by parameters $\boldsymbol{\theta}_i$, transforms data from the original input domain $\mathbb{R}^D$ into a feature space $\mathbb{R}^d$. Its classifier, represented by $g_i$ with parameters $\boldsymbol{\phi}_i$, then maps these features to the final output space $\mathbb{R}^K$.

**Local Prototype** The local prototype of class $j$ on client $i$, denoted by $\bar{\boldsymbol{c}}_{i,j}^L$, is defined as the mean of the feature embedding vectors of samples from class $j$ in client $i$'s local dataset. Formally,

$$\bar{\boldsymbol{c}}_{i,j}^L = \frac{1}{n_{i,j}} \sum_{(x,y) \in \mathcal{D}_{i,j}} f_i(\boldsymbol{\theta}_i; x), \tag{3}$$

where $n_{i,j} = |\mathcal{D}_{i,j}|$ is the number of samples from class $j$ on client $i$, $\mathcal{D}_{i,j} \subseteq \mathcal{D}_i$ is the subset of client $i$'s local dataset containing samples from class $j$.

**Global Prototype** The global prototype of class $j$ can be defined as an average of the local prototypes. A simple averaging method without weighting is given by:

$$\bar{\boldsymbol{c}}_j^G = \frac{1}{|\mathcal{N}_j|} \sum_{i \in \mathcal{N}_j} \bar{\boldsymbol{c}}_{i,j}^L, \tag{4}$$

where $\mathcal{N}_j$ represents the set of clients with samples from class $j$.

**Training Objective of PBFL** PBFL optimizes a combined loss function comprising a supervised learning loss and a regularization term that minimizes the distance between local and global prototypes. The total loss for client $i$ is defined as:

$$\tilde{\mathcal{L}}_i(\boldsymbol{w}_i) = \mathcal{L}_i(\boldsymbol{w}_i) + \lambda \Omega_i, \tag{5}$$

where $\Omega_i$ is the regularization term and $\lambda$ is a hyperparameter controlling regularization strength. The term $\Omega_i$ is formulated as:

$$\Omega_i = \sum_j \rho(\bar{\boldsymbol{c}}_{i,j}^L, \bar{\boldsymbol{c}}_j^G), \tag{6}$$

where the function $\rho(\cdot, \cdot)$ computes the Euclidean distance between the two prototypes.

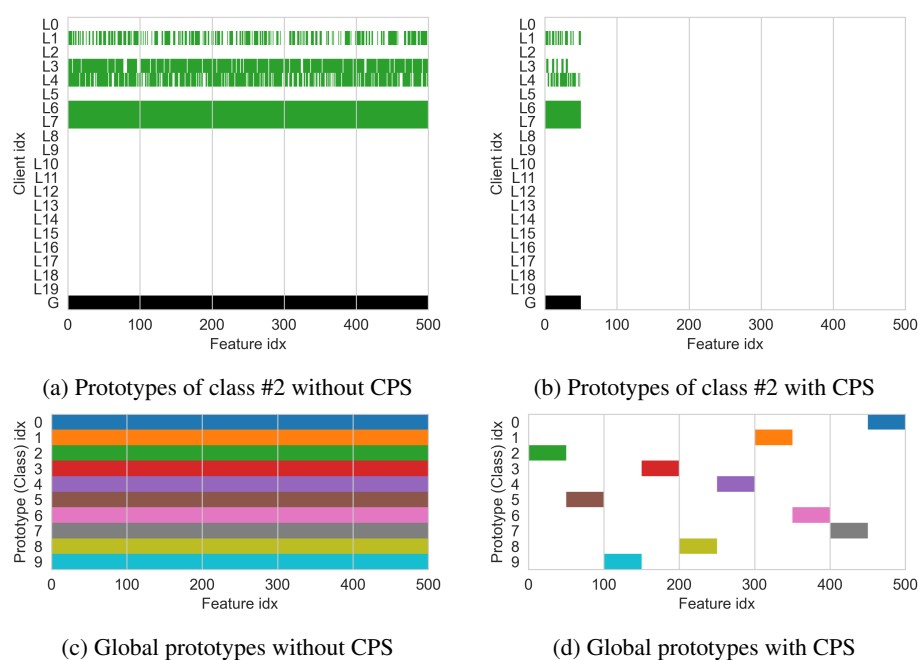

(a) Prototypes of class #2 without CPS  (b) Prototypes of class #2 with CPS

(c) Global prototypes without CPS  (d) Global prototypes with CPS

Figure 1: Prototype comparison of FedProto with and without CPS for the CIFAR-10 dataset. Each row in the heatmaps represents a prototype, and a colored cell indicates a non-zero value. The dimension $s$ is 50. More examples are provided in the appendix.

## 4 METHODS

This section provides a comprehensive overview of our three proposed methods.

### 4.1 ADAPTATION OF PROTOTYPES WITH STRUCTURED SPARSITY

Representation layers often exhibit sparsity when using the ReLU (Rectified Linear Unit) activation function, which can lead to 'dead' hidden units. A *dead unit* is defined as a hidden unit that outputs zero for all input patterns in the training set (Lu et al., 2019), effectively not contributing to learning or inference. Our observations reveal that in the decision layer of a deep network, nearly half of the hidden units can be dead per class. Figure 1a illustrates this phenomenon, displaying a heatmap of 500-dimensional local prototypes for 20 clients (L0-L19) and the global prototype for class #2 of the CIFAR-10 dataset after completing FL with FedProto. In this visualization, colored features indicate non-zero values, while blank areas represent zeros (dead units). Notably, some clients show zero prototype vectors, indicating the absence of class #2 in their local dataset. Several clients (L1, L3, L4) utilize only partial feature dimensions.

Intriguingly, despite these sparse representations, deep networks maintain high performance. This resilience can be attributed to the networks' substantial capacity and robust generalization capabilities (Arpit et al., 2017; Zhang et al., 2021a; Kawaguchi et al., 2022). Given these observations, one might hypothesize that the sparsity in a decision layer (prototype, feature embedding) could be advantageous, potentially reducing communication requirements between the server and clients if the sparse locations were consistent across clients. However, in PBFL scenarios, the locations of dead units typically vary among clients, as evident in Figure 1a due to the heterogeneity of models and data across clients.

**Class-wise Prototype Sparsification (CPS)** To leverage sparsity benefits in PBFL, we propose Class-wise Prototype Sparsification (CPS). This method imposes structured sparsity per class, ensuring consistency in zero locations across clients. CPS implementation is straightforward, involving sharing predetermined sparse locations in prototypes. We introduce class-specific binary masking vectors $\boldsymbol{m}_j \in \{0,1\}^d$, which determine which prototype vector elements are set to zero, creating a 'structured sparse prototype.' We omit superscripts and subscripts for simplicity, representing a

masking vector as $\boldsymbol{m}$ and a prototype as $\bar{c}$. Let $\boldsymbol{m} = (m_1, m_2, \ldots, m_d) \in \{0, 1\}^d$ be a masking vector and $\bar{\boldsymbol{c}} = (\bar{c}_1, \bar{c}_2, \ldots, \bar{c}_d) \in \mathbb{R}^d$ be a prototype. With $\boldsymbol{m}$ and $\bar{\boldsymbol{c}}$, we define the structured sparse prototype for updating local models and the compressed prototype for communicating prototypes.

**Definition 1** (Structured Sparse Prototype). *The structured sparse prototype $\tilde{\boldsymbol{c}} \in \mathbb{R}^d$ is defined as:*

$$\tilde{\boldsymbol{c}} = \boldsymbol{m} \odot \bar{\boldsymbol{c}}, \tag{7}$$

*where $\odot$ denotes the Hadamard product.*

**Definition 2** (Compressed Prototype). *The compressed prototype $\hat{\boldsymbol{c}} \in \mathbb{R}^s$ is defined as:*

$$\hat{\boldsymbol{c}} = (\bar{c}_i : m_i = 1), \tag{8}$$

*where $s = \sum_{i=1}^d m_i$ is the number of non-zero elements in $\boldsymbol{m}$.*

Sharing $\boldsymbol{m}$ between the server and clients allows for efficient communication. Instead of transmitting the complete prototype $\bar{\boldsymbol{c}}$, only the compressed prototype $\hat{\boldsymbol{c}}$ needs to be communicated. This $\hat{\boldsymbol{c}}$ contains only the non-zero elements specified by $\boldsymbol{m}$, as illustrated by the colored dimensions in Figures 1b and 1d.

To reduce the communication cost of sending $\boldsymbol{m}$, the $d$-dimensional $\boldsymbol{m}$ can be mapped to a $K$-dimensional vector, where each element represents $\frac{d}{K}$ consecutive dimensions of $\boldsymbol{m}$ for each class. For instance, with $K = 10$ and $d = 500$, each prototype is allocated a block of 50 consecutive dimensions (Figure 1d). We typically maximize pairwise Hamming distances between the $K$ vectors to ensure inter-class distinctiveness.

## 4.2 Aggregation of Local Prototypes without Using Local Data Distribution

One commonly used aggregation method, as described in (Tan et al., 2022a; Zhang et al., 2024), computes the global prototype for class $j$ using a weighted average of the local prototypes:

$$\bar{\boldsymbol{c}}_j^G = \frac{1}{|\mathcal{N}_j|} \sum_{i \in \mathcal{N}_j} \frac{n_{i,j}}{\sum_{i=1}^M n_{i,j}} \bar{\boldsymbol{c}}_{i,j}^L, \tag{9}$$

where $\sum_{i=1}^M n_{i,j}$ denotes the number of class $j$-th samples across all clients. The weighting factor $\frac{n_{i,j}}{\sum_{i=1}^M n_{i,j}}$ ensures that each local prototype's contribution to the global prototype is proportional to the number of samples from class $j$ on the corresponding client among all samples from class $j$. The normalization factor $\frac{1}{|\mathcal{N}_j|}$ ensures scaling of the global prototype. However, this aggregation method can potentially violate privacy in some applications due to the requirement for the server to receive information about clients' local data distribution. Specifically, the server needs to know the number of samples from class $j$ on client $i$, which can pose privacy risks in many FL applications."

**Privacy-preserving Prototype Aggregation (PPA)** To address the privacy concerns inherent in the aggregation method (Eq. (9)), we propose Privacy-preserving Prototype Aggregation (PPA). This method enhances data protection by modifying the aggregation technique as follows:

$$\bar{\boldsymbol{c}}_j^G = K \sum_{i \in \mathcal{N}_j} \frac{n_i}{n} \frac{n_{i,j}}{n_i} \bar{\boldsymbol{c}}_{i,j}^L \tag{10}$$

$$= \frac{K}{n} \sum_{i \in \mathcal{N}_j} n_{i,j} \bar{\boldsymbol{c}}_{i,j}^L, \tag{11}$$

where $\frac{n_i}{n}$ represents the proportion of samples on client $i$ relative to the total samples across all clients, $\frac{n_{i,j}}{n_i}$ denotes the proportion of samples from class $j$ on client $i$ relative to the total samples on that client, and $K$ is a normalization factor ensuring proper scaling of $\bar{\boldsymbol{c}}_j^G$. These ratios effectively capture the overall contribution of client $i$ to the system and the prevalence of class $j$ within that client's dataset. The PPA method offers enhanced privacy protection compared to Eq. (9). In Eq. (11), only $K$ and $n$ are known to the server and remain constant across all clients. This design allows each client to transmit only the product $n_{i,j} \bar{\boldsymbol{c}}_{i,j}^L$, with the server performing the final scaling by $\frac{K}{n}$. By construction, $\bar{\boldsymbol{c}}_j^G$ and $\bar{\boldsymbol{c}}_{i,j}^L$ can be replaced with compressed prototypes $\hat{\boldsymbol{c}}_j^G$ and $\hat{\boldsymbol{c}}_{i,j}^L$.

Notably, under certain conditions, PPA exhibits close relationships with other methods.

**Remark 1.** *Consider a scenario where all clients have samples from all classes, with an equal number of samples across clients and a uniform class distribution. Under these conditions, two relationships emerge. First, the PPA method, as defined in Eq. (11), is equivalent to the simple averaging method in Eq. (4). Second, the PPA method becomes equivalent to the weighted-averaging method in Eq. (9), scaled by a factor of $\frac{1}{|\mathcal{N}_j|}$.*

Detailed explanations and derivations for these relationships are provided in the appendix.

### 4.3 DISTILLATION FROM GLOBAL PROTOTYPES WITH LOCAL DATA DISTRIBUTION

In PBFL, personalization is achieved by allowing local models to learn exclusively from global prototypes corresponding to classes in their local datasets. The strength of this knowledge distillation is regulated by a single hyperparameter $\lambda$. However, this approach can still distill from undistillable classes (Zhu et al., 2022), which means that it may not adequately prioritize learning from global prototypes of classes that are more prevalent in the client's local dataset while potentially over-emphasizing less common classes.

**Class-Proportional Knowledge Distillation (CPKD)** To enhance the utilization of class-specific global prototypes, we propose a weighted distillation approach that accounts for the class distribution in each client's local dataset. The Class-Proportional Knowledge Distillation (CPKD) method introduces a weight term $\beta$ to adjust the distillation strength for each global prototype. Specifically, we modify $\Omega_i$ as follows:

$$\Omega_i = \sum_j \beta_{i,j} \rho(\bar{c}_{i,j}^L, \bar{c}_j^G), \tag{12}$$

where $\beta_{i,j} = \frac{p_{i,j}}{\max_k(p_{i,k})}$ represents a class-specific weight for client $i$ and class $j$. In this formulation, $p_{i,j}$ denotes the proportion of samples from class $j$ in client $i$'s dataset, calculated as $p_{i,j} = \frac{n_{i,j}}{n_i}$. By defining $\beta_{i,j}$ in this manner, we ensure that the weight is proportional to the empirical class distribution of the local dataset. When combining CPS with CPKD, we replace $\bar{c}_j^G$ and $\bar{c}_{i,j}^L$ with their structured sparse counterparts $\tilde{c}_j^G$ and $\tilde{c}_{i,j}^L$, respectively.

### 4.4 INTEGRATION OF PROPOSED METHODS INTO PBFL

The proposed methods' strength lies in their seamless integration into existing PBFL algorithms. When incorporating CPS into vanilla PBFL (FedProto), we need to make modifications such as creating and sharing masking vectors and sparsifying and reconstructing prototypes using these vectors. Similarly, PPA and CPKD can be applied to FedProto by replacing its aggregation and distillation parts. These components can be integrated into other PBFL algorithms, such as FedTGP. A detailed algorithm is provided in the appendix.

## 5 EXPERIMENTS

In this section, we evaluate the performance and communication efficiency of our proposed methods and analyze the impact of incorporating CPS and CPKD techniques into PBFL approaches.

### 5.1 EXPERIMENTAL SETUP

We utilize three datasets to evaluate federated learning algorithms: CIFAR-10, CIFAR-100 (Krizhevsky et al., 2009), and TinyImageNet (Le & Yang, 2015). Each dataset is partitioned into training (75%) and test (25%) sets. We simulate real-world federated learning scenarios by creating heterogeneous data distributions across clients using a Dirichlet distribution ($\text{Dir}(\alpha)$) with $\alpha$ set to 0.1 by default (Lin et al., 2020). For our experiments, we employ four lightweight models suitable for resource-constrained devices: ResNet8 (Zhong et al., 2017), EfficientNet (Tan, 2019), ShuffleNetV2 (Ma et al., 2018), and MobileNetV2 (Sandler et al., 2018). Each model incorporates a global average pooling layer (Szegedy et al., 2015), setting the prototype dimension $d = 500$.

Our federated learning environment comprises 20 clients, all actively participating in each of the 300 communication rounds. The client-side configuration includes a learning rate of 0.01, a batch

Table 1: Classification accuracy (Acc.) and communication cost (Comm.) across datasets. The CPS column shows compressed prototype dimension $s$. The mark ✓ indicates the method used. Comm. is measured by the number of parameters shared per FL round. 'M' is short for million.

| Algorithm | Our method | | | CIFAR-10 | | CIFAR-100 | | TinyImageNet | |
|---|---|---|---|---|---|---|---|---|---|
| | CPS | PPA | CPKD | Acc. (%) | Comm. | Acc. (%) | Comm. | Acc. (%) | Comm. |
| LG-FedAvg | | | | $86.91 \pm 0.14$ | 0.20M | $38.54 \pm 0.21$ | 2.00M | $22.30 \pm 0.37$ | 4.00M |
| FML | | | | $86.59 \pm 0.15$ | 34.32M | $37.83 \pm 0.03$ | 36.12M | $22.03 \pm 0.12$ | 38.12M |
| FedKD | | | | $87.10 \pm 0.02$ | 30.66M | $39.74 \pm 0.42$ | 32.26M | $23.08 \pm 0.17$ | 34.05M |
| FedDistill | | | | $86.93 \pm 0.12$ | <0.01M | $39.52 \pm 0.33$ | 0.29M | $22.98 \pm 0.15$ | 1.17M |
| FedProto | | | | $82.90 \pm 0.46$ | 0.15M | $29.97 \pm 0.18$ | 1.46M | $13.30 \pm 0.06$ | 2.93M |
| FedProto | 50 | | | $84.18 \pm 0.71$ | 0.02M | $29.27 \pm 0.28$ | 0.15M | $10.02 \pm 0.24$ | 0.29M |
| FedProto | 250 | | | $84.30 \pm 0.16$ | 0.08M | $33.00 \pm 0.28$ | 0.73M | $15.70 \pm 0.47$ | 1.46M |
| FedProto | | ✓ | | $84.89 \pm 0.29$ | 0.15M | $34.63 \pm 0.10$ | 1.46M | $19.19 \pm 0.09$ | 2.93M |
| FedProto | | | ✓ | $85.18 \pm 0.09$ | 0.15M | $33.03 \pm 0.49$ | 1.46M | $11.43 \pm 0.22$ | 2.93M |
| FedProto | 50 | ✓ | ✓ | $85.54 \pm 0.54$ | 0.02M | $34.76 \pm 0.22$ | 0.15M | $18.49 \pm 0.11$ | 0.29M |
| FedProto | 250 | ✓ | ✓ | $85.23 \pm 0.51$ | 0.08M | $37.82 \pm 0.25$ | 0.73M | $21.41 \pm 0.22$ | 1.46M |
| FedTGP | | | | $86.32 \pm 0.49$ | 0.15M | $36.92 \pm 0.16$ | 1.46M | $19.44 \pm 0.12$ | 2.93M |
| FedTGP | 50 | | | $85.72 \pm 0.37$ | 0.02M | $34.72 \pm 0.85$ | 0.15M | $16.35 \pm 0.46$ | 0.29M |
| FedTGP | 250 | | | $85.84 \pm 0.13$ | 0.08M | $34.27 \pm 0.60$ | 0.73M | $17.30 \pm 0.35$ | 1.46M |
| FedTGP | | ✓ | | $\mathbf{87.65 \pm 0.34}$ | $\mathbf{0.15M}$ | $\mathbf{45.84 \pm 0.65}$ | $\mathbf{1.46M}$ | $26.90 \pm 0.28$ | 2.93M |
| FedTGP | | | ✓ | $87.18 \pm 0.14$ | 0.15M | $39.10 \pm 0.11$ | 1.46M | $22.31 \pm 0.13$ | 2.93M |
| FedTGP | 50 | ✓ | ✓ | $\underline{87.11 \pm 0.08}$ | $\underline{0.02M}$ | $\underline{43.64 \pm 0.29}$ | $\underline{0.15M}$ | $\mathbf{27.82 \pm 0.23}$ | $\mathbf{0.29M}$ |
| FedTGP | 250 | ✓ | ✓ | $87.20 \pm 0.21$ | 0.08M | $43.21 \pm 0.59$ | 0.73M | $25.92 \pm 0.39$ | 1.46M |

size of 32, and 1 local training epoch per round. We evaluate our proposed methods, integrated with FedProto and FedTGP, against four data-free federated learning algorithms: LG-FedAVG (Liang et al., 2020), FML (Shen et al., 2020), FedKD (Zhu et al., 2021), and FedDistill (Jeong et al., 2018). For FedProto and FedTGP, we calculate accuracy based on the L2 distance between each sample's representational vector $f(\theta; x)$ and the global class prototypes $\bar{\mathbf{c}}_j^G$, as described in (Tan et al., 2022a). We set the hyperparameters for these methods according to their original papers: $\lambda = 0.1$ (prototype loss regularizer), $\tau = 100$ (margin threshold), and $S = 100$ (prototype training epoch).

Our primary evaluation metric is the highest mean test accuracy achieved by each algorithm across all communication rounds, a widely adopted measure in federated learning literature (McMahan et al., 2017). We report the average results from three independent experiments conducted with different random seeds to ensure statistical robustness. For fairness, we apply no hyperparameter schedulers during training. Detailed information regarding the experimental setup and additional configurations is provided in the appendix to ensure reproducibility.

## 5.2 COMPARISON OF PERFORMANCE AND COMMUNICATION COST

**Performance Improvement**    Table 1 demonstrates the efficacy of our proposed methods when integrated with FedProto and FedTGP. Notably, FedTGP combined with our approaches consistently outperforms various algorithms across different settings, as highlighted in **bold**. In particular, FedTGP paired with PPA alone exhibits significant performance gains. This improvement likely stems from the approach's ability to effectively incorporate the relative importance of each client's local prototype during the contrastive learning process in the server.

**Communication Cost Reduction**    To investigate communication cost, we experimented with varying the dimensions of the compressed prototype $s$. FedTGP with the three methods at $s = 50$ surpasses all baseline algorithms in performance (underlined). This configuration achieves up to a 4x reduction in communication costs compared to FedDistill, the most communication-efficient approach among the baselines. While FedDistill demonstrates the lowest communication cost for CIFAR-10 due to the smaller number of classes relative to prototype dimensions, it is important to note that transmitting averaged logits can pose higher privacy risks than transmitting prototypes.

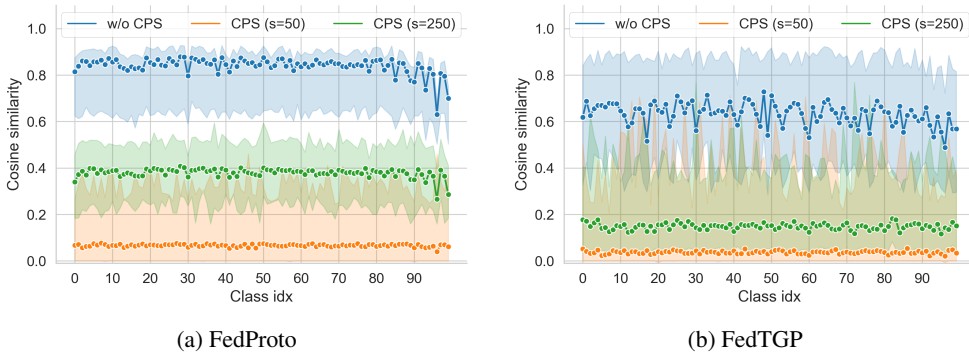

(a) FedProto  (b) FedTGP

Figure 2: Cosine similarity comparison of global prototypes with and without CPS. The dimension of compressed prototype $s$ is set to 50 or 250.

Table 2: Classification accuracy for different dimensions of feature space and compressed prototypes. 'Feature control' refers to controlling the number of neurons $d$ in the decision layer. 'CPS' indicates that CPS is applied with the dimension of the compressed prototype $s$.

| Dimension | CIFAR-10 | | CIFAR-100 | |
|---|---|---|---|---|
| | CPS ($s$) | Feature control ($d$) | CPS ($s$) | Feature control ($d$) |
| 50 | $84.18 \pm 0.71$ | $79.05 \pm 0.56$ | $29.27 \pm 0.18$ | $24.94 \pm 0.55$ |
| 150 | $84.25 \pm 0.23$ | $80.58 \pm 1.30$ | $32.43 \pm 0.52$ | $29.82 \pm 0.43$ |
| 250 | $84.30 \pm 0.16$ | $82.27 \pm 1.09$ | $33.00 \pm 0.28$ | $30.41 \pm 0.28$ |
| 350 | $84.07 \pm 0.50$ | $82.46 \pm 1.38$ | $32.86 \pm 0.34$ | $30.84 \pm 0.15$ |
| 450 | $83.32 \pm 0.18$ | $82.98 \pm 0.91$ | $31.58 \pm 0.42$ | $31.31 \pm 0.22$ |

**Ablation Test**  Our ablation studies demonstrate that combining FedProto or FedTGP with any individual proposed method generally yields performance improvements over the original approaches. However, we observe an exception in the case of FedTGP combined with CPS alone, which fails to show enhancement. Notably, no individual method consistently excels in all scenarios, and we verified that no combination of two methods outperforms the integration of all three. These findings imply that the best combination of methods might depend on the specific characteristics of the considered federated learning environment.

### 5.3 ANALYSIS ON EFFECT OF CPS AND CPKD

**Distance between Global Prototypes**  We analyze the pair-wise cosine similarities of global prototypes under the application of CPS, as illustrated in Figure 2. Cosine similarity is our chosen distance metric for global prototypes due to its invariance to vector scaling. The line plots in Figure 2 depict the average cosine similarity between each global prototype and all others, with the shaded regions indicating the range between maximum and minimum similarities. As compression levels increase ($s$ decreases), cosine similarity values decrease as expected. FedTGP consistently shows lower cosine similarity than FedProto across all compression levels. This finding aligns with Zhang et al. (2024)'s suggestion that higher prototype distinctiveness contributes to improved performance.

To investigate the effectiveness of the CPS method, we conducted experiments varying the dimension of the compressed prototype $s$ for FedProto. Table 2 presents the performance changes corresponding to different compressed prototype dimensions. Our results reveal a clear trade-off between dimensionality (affecting communication cost) and classification accuracy. Notably, a dimensionality of 250 achieved the optimal balance, demonstrating the best overall performance. To provide context for these results, we compared CPS with an alternative approach that reduced the number of hidden neurons $d$ in the decision layer—this alternative method aimed to achieve the same communication cost as CPS. The comparison shows that CPS consistently outperforms the reduced hidden neuron approach. This superiority stems from CPS's ability to fully utilize a larger decision layer dimension during model training.

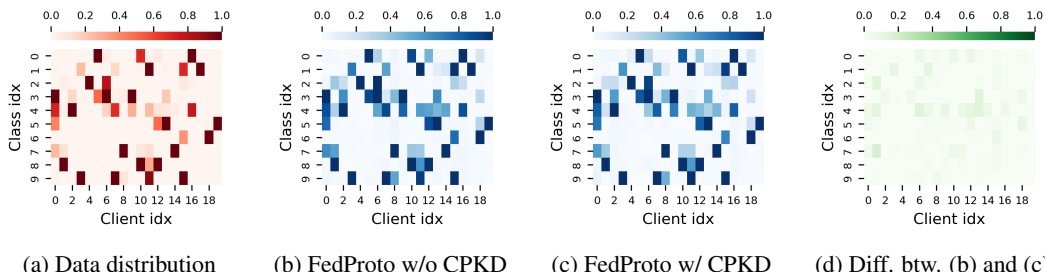

(a) Data distribution     (b) FedProto w/o CPKD     (c) FedProto w/ CPKD     (d) Diff. btw. (b) and (c)

Figure 3: Heatmaps depicting the data distribution and L2-norms of class-specific weight vectors for CIFAR-10. (a) Each cell represents the normalized number of data samples belonging to class $j$ for client $i$. (b)-(d) Each cell shows normalized $\|\phi_{i,j}\|_2$ of models.

**Personalization by CPKD** We have previously confirmed the enhanced personalization through improved accuracies, as shown in Table 1. To further validate this finding, we now employ a visualization method proposed by Lee & Choi (2024). This method infers the degree of personalization by comparing the local data's class distribution with the weight distribution of a deep network. We denote the set of weights connecting the decision layer to the output layer for client $i$ as $\phi_i = (\phi_{i,1}, \phi_{i,2}, ..., \phi_{i,K})$. Here, each $\phi_{i,j}$ represents the weight vector linking the decision layer's hidden units to the output unit corresponding to class $j$. The method is based on the observed correlation between the L2-norm distribution of these $\phi_{i,j}$ vectors and the client's local class distribution. The authors propose an approximate relationship $\frac{\mathbb{E}\|\phi_{i,j}\|_2^2}{\mathbb{E}\|\phi_{i,k}\|_2^2} \approx \frac{n_{i,j}^2}{n_{i,k}^2}$. This relationship draws its foundation from the work of (Anand et al., 1993), which established a correlation between the gradient of $\phi_{i,j}$ and local dataset class distributions.

To demonstrate how CPKD enhances personalization, we visualize heatmaps in Figure 3. These heatmaps depict normalized values of the data distribution across clients and the L2-norms of weight vectors ($\|\phi_{i,j}\|_2$) for local models. We normalized the values from 0 to 1 using column-wise min-max normalization for the heatmaps. Examination of the heatmaps for FedProto, both with and without CPKD (Figures 3b and 3c), reveals patterns similar to the data distribution heatmap (Figure 3a), indicating effective personalization. However, a closer inspection reveals subtle differences between these FedProto heatmaps (Figure 3d). We calculate the Frobenius norm of the difference between the data distribution heatmap and each FedProto heatmap to quantify these differences. This calculation yields a value of 2.14 for FedProto without CPKD and 1.73 for FedProto with CPKD. The lower Frobenius norm for FedProto with CPKD indicates that its heatmap more closely aligns with the data distribution than FedProto without CPKD. This result suggests that CPKD indeed enhances the personalization of local models. We corroborate this finding with similar results from our analysis of the CIFAR-100 dataset, as presented in the appendix.

## 6 DISCUSSION AND CONCLUSION

Our study presents quantitative evidence on the effects of three methods on FedProto and FedTGP. However, the mechanisms by which these methods operate remain partially unclear. For instance, while applying CPS alone to FedProto generally improves accuracy, it does not yield similar benefits for FedTGP (Table 1). We investigated the effect of sparsity proportion but did not observe a consistent relationship between the cosine similarities of global prototypes and performance (Figure 2, Tables 1 and 2). This lack of consistency suggests a trade-off between prototype distances and deep network capacity for efficient training on different sizes of decision layers.

Our evaluation results indicate that while PBFL approaches offer advantages regarding communication cost, privacy, and personalization, they tend to underperform compared to several existing data-free FL approaches when used alone. However, by incorporating our methods, which can be easily integrated into each stage of PBFL, we have enhanced the original PBFL approaches. In resource-constrained environments, these enhanced PBFL approaches may prove more practical.

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

# APPENDIX

## A  VISUALIZATION OF STRUCTURED SPARSE PROTOTYPES

In this section, we provide visualizations of structured sparse prototypes in both their original and binary forms for structured sparse prototype dimension sparsity levels (s) and datasets. The heatmaps are presented in pairs: those on the left depict the original values of the prototypes, while those on the right show the same prototypes with values converted to 1 when larger than 0, and 0 otherwise.

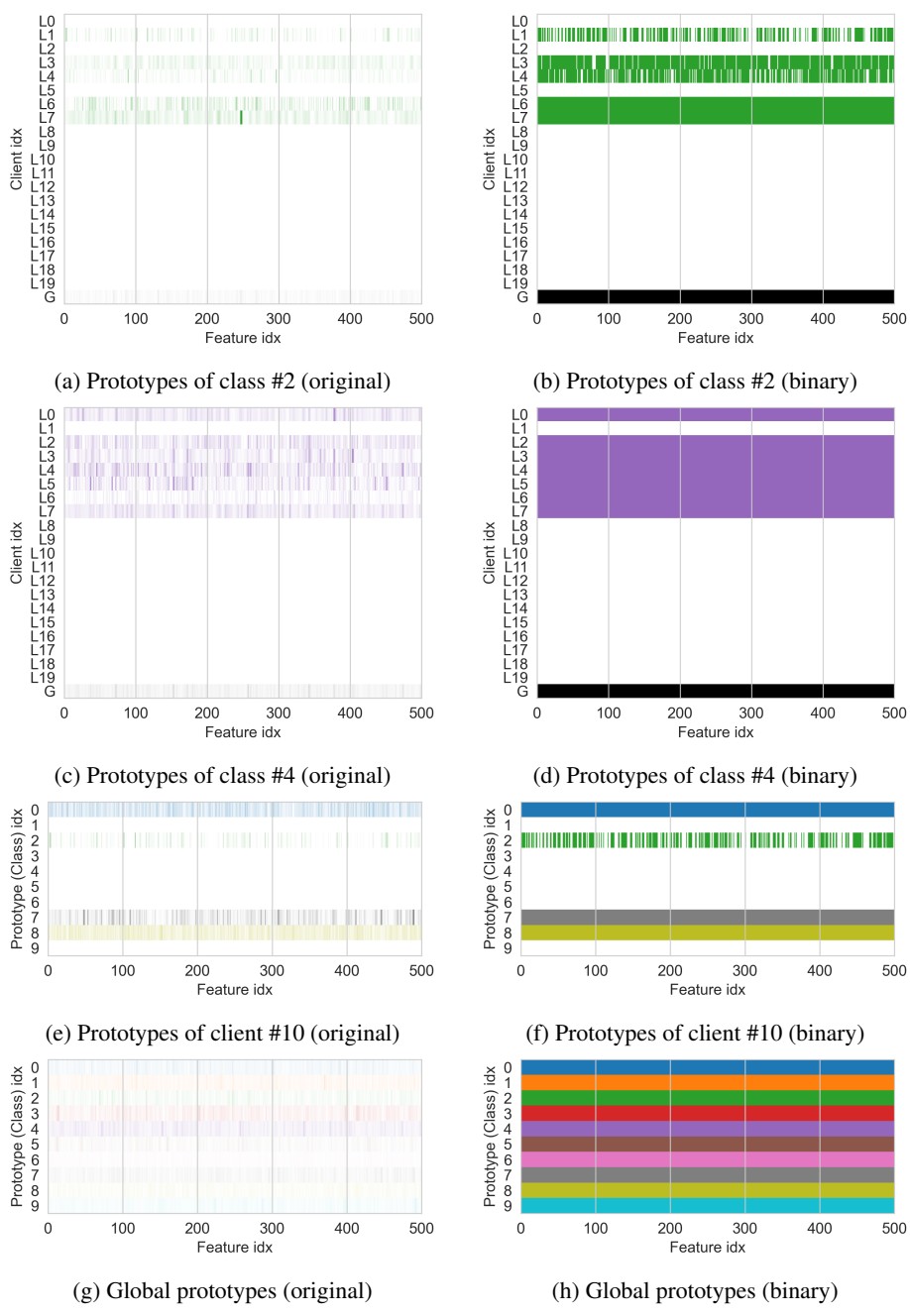

(a) Prototypes of class #2 (original)     (b) Prototypes of class #2 (binary)

(c) Prototypes of class #4 (original)     (d) Prototypes of class #4 (binary)

(e) Prototypes of client #10 (original)     (f) Prototypes of client #10 (binary)

(g) Global prototypes (original)     (h) Global prototypes (binary)

Figure 4: Prototype comparison of FedProto without Class-wise Prototype Sparsification (CPS) for the CIFAR-10 dataset.

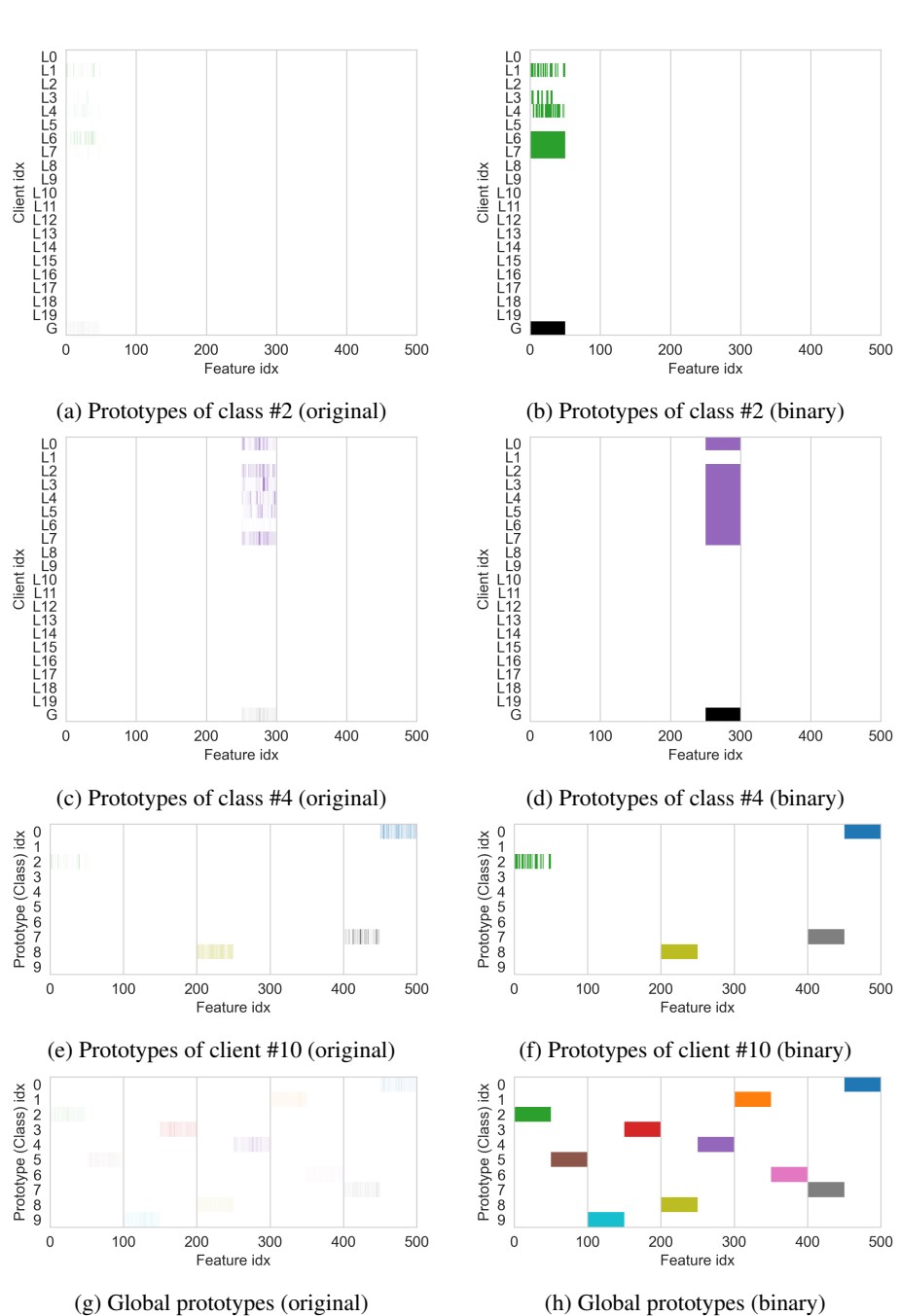

(a) Prototypes of class #2 (original)

(b) Prototypes of class #2 (binary)

(c) Prototypes of class #4 (original)

(d) Prototypes of class #4 (binary)

(e) Prototypes of client #10 (original)

(f) Prototypes of client #10 (binary)

(g) Global prototypes (original)

(h) Global prototypes (binary)

Figure 5: Prototype comparison of FedProto with Class-wise Prototype Sparsification (CPS) for the CIFAR-10 dataset. The dimension $s$ is 50 for CPS.

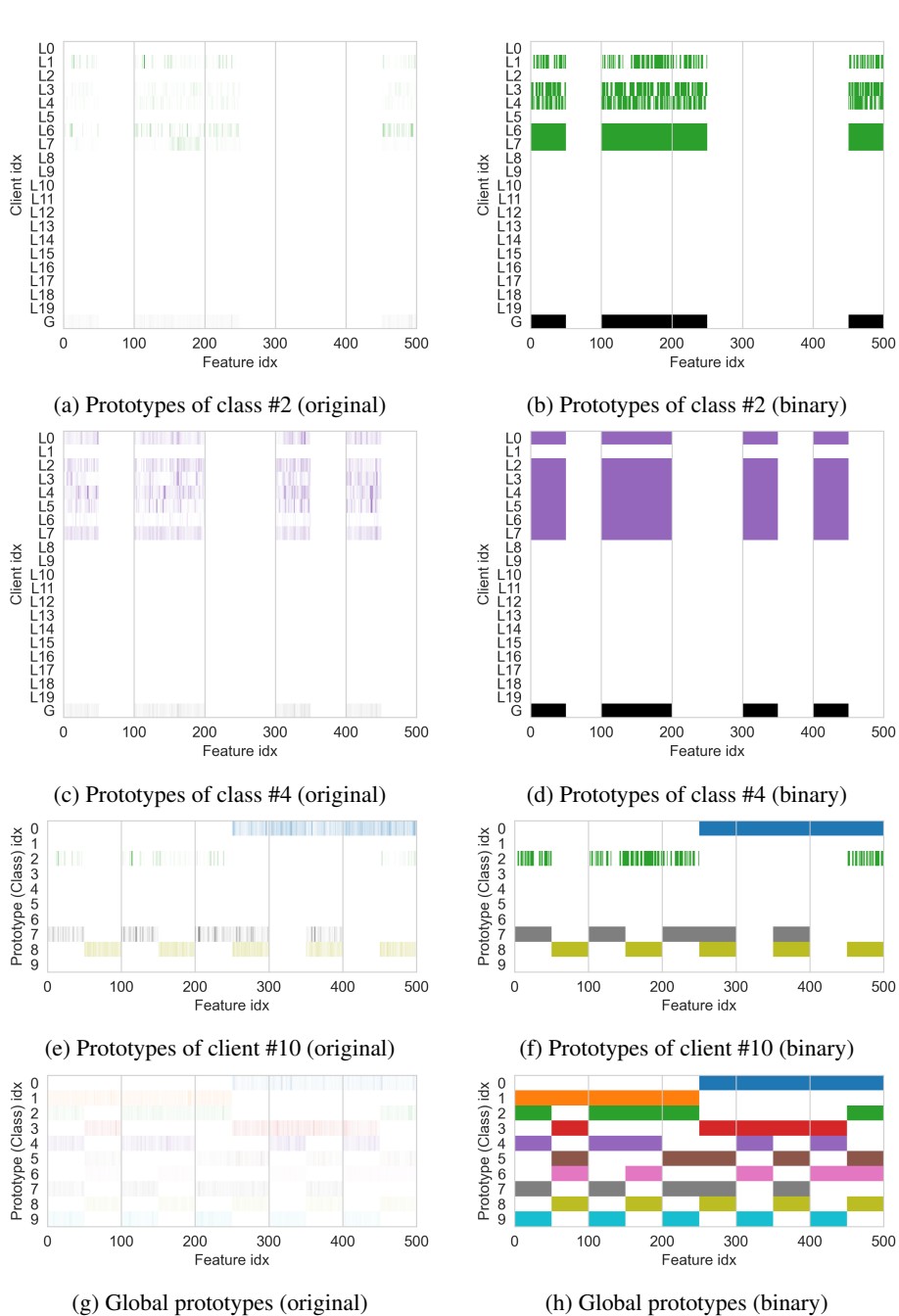

(a) Prototypes of class #2 (original)   (b) Prototypes of class #2 (binary)

(c) Prototypes of class #4 (original)   (d) Prototypes of class #4 (binary)

(e) Prototypes of client #10 (original)   (f) Prototypes of client #10 (binary)

(g) Global prototypes (original)   (h) Global prototypes (binary)

Figure 6: Prototype comparison of FedProto with Class-wise Prototype Sparsification (CPS) for the CIFAR-10 dataset. The dimension $s$ is 250 for CPS.

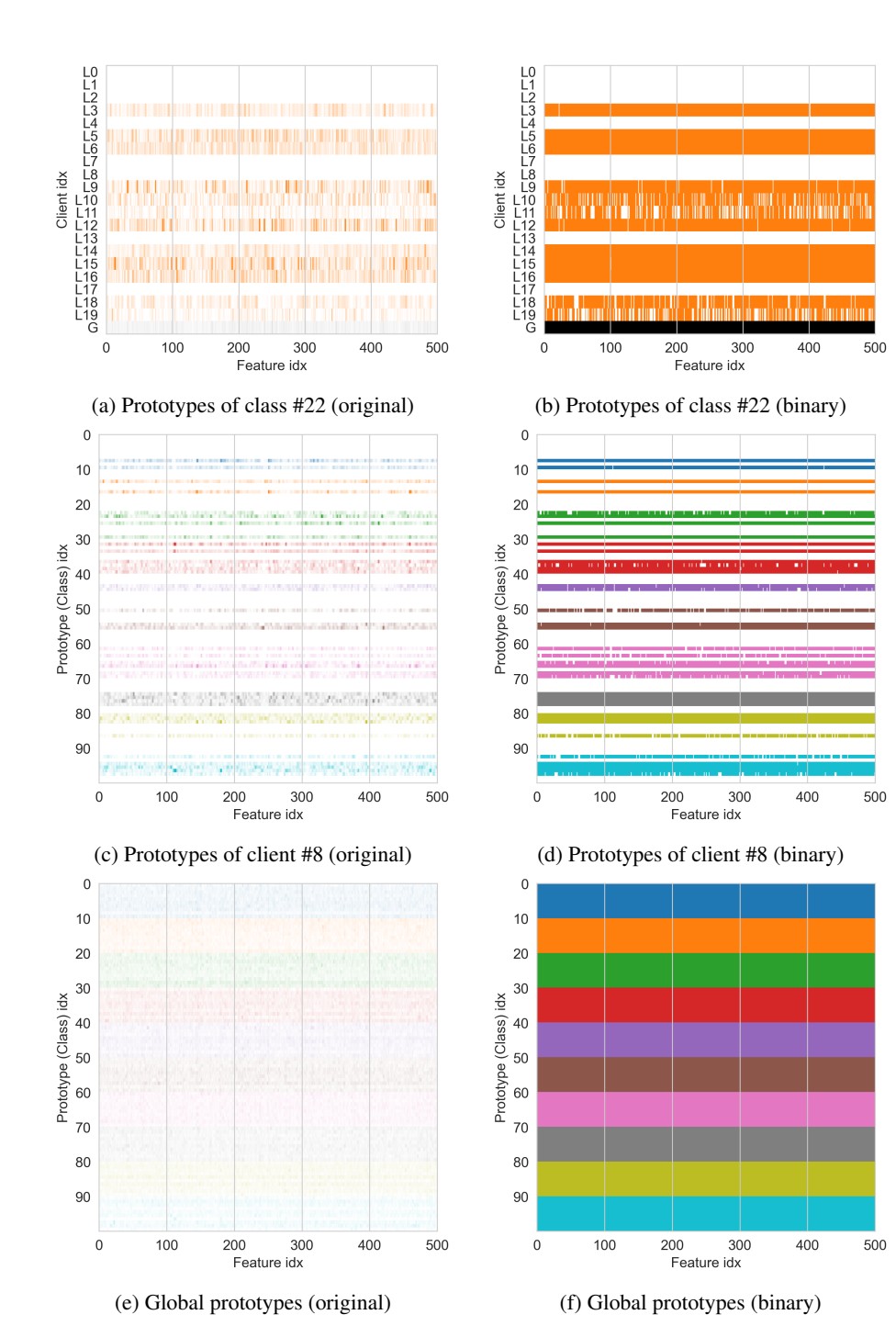

(a) Prototypes of class #22 (original)

(b) Prototypes of class #22 (binary)

(c) Prototypes of client #8 (original)

(d) Prototypes of client #8 (binary)

(e) Global prototypes (original)

(f) Global prototypes (binary)

Figure 7: Prototype comparison of FedProto without Class-wise Prototype Sparsification (CPS) for the CIFAR-100 dataset.

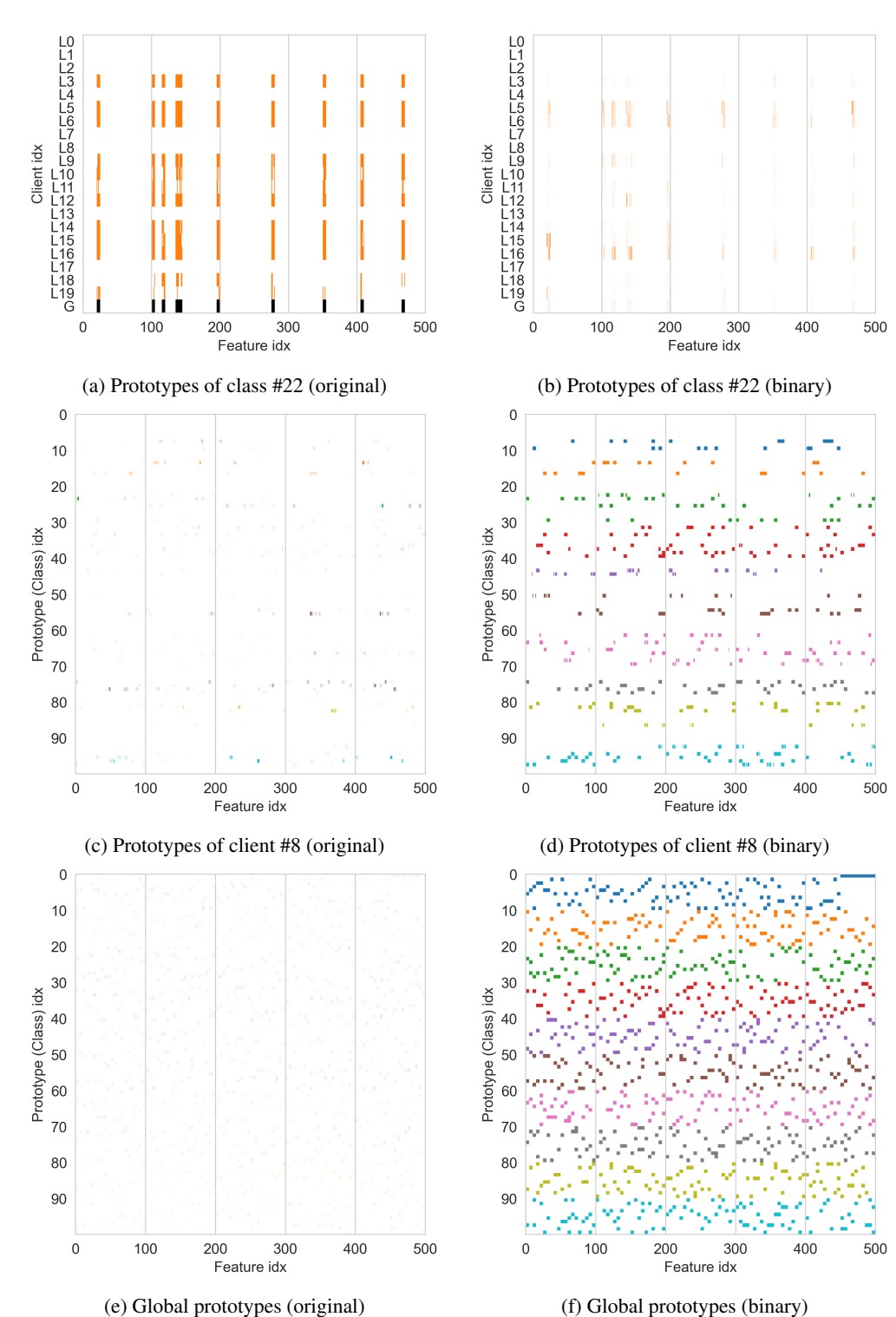

Figure 8: Prototype comparison of FedProto with Class-wise Prototype Sparsification (CPS) for the CIFAR-100 dataset. The dimension $s$ is 50 for CPS.

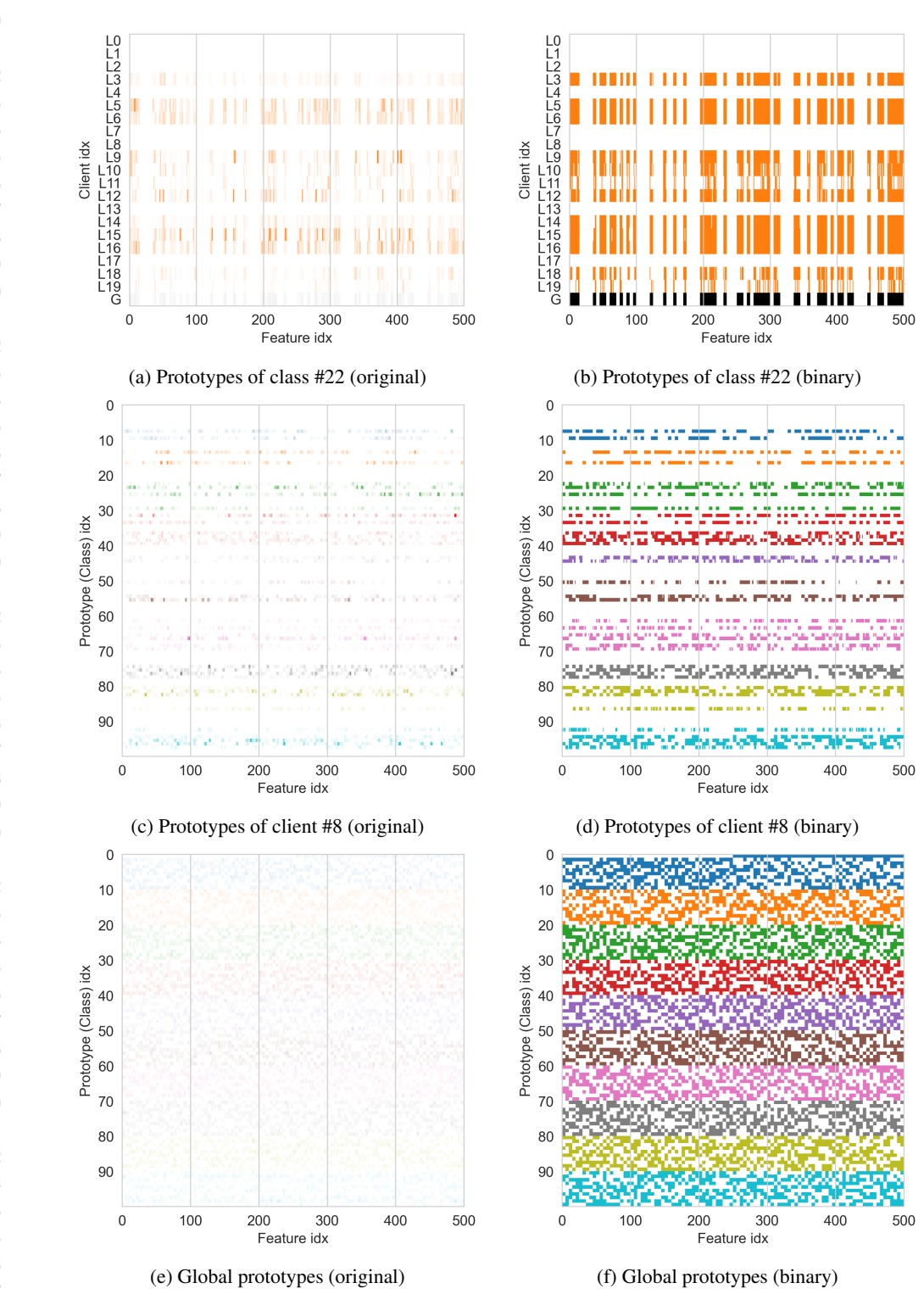

(a) Prototypes of class #22 (original)

(b) Prototypes of class #22 (binary)

(c) Prototypes of client #8 (original)

(d) Prototypes of client #8 (binary)

(e) Global prototypes (original)

(f) Global prototypes (binary)

Figure 9: Prototype comparison of FedProto with Class-wise Prototype Sparsification (CPS) for the CIFAR-100 dataset. The dimension $s$ is 250 for CPS.

## B   Detailed Explanation for Remark 1

In this section, we elucidate the relationships described in Remark 1. Specifically, we demonstrate how the Privacy-Preserving Prototype Aggregation (PPA) method, as defined in Eq. (11), relates to two other aggregation methods: the simple aggregation method presented in Eq. (4) and the weighted-averaging method shown in Eq. (9). These relationships emerge under specific conditions and provide insights into the behavior of the PPA method.

### B.1   Relationship between PPA and the Simple Averaging Method

As noted in Remark 1, under specific conditions, the PPA method (Eq. (11)) becomes identical to the simple averaging method (Eq. (4)). We can demonstrate this equivalence through the following derivation:

$$\bar{c}_j^G = \frac{K}{n} \sum_{i \in \mathcal{N}_j} n_{i,j} \bar{c}_{i,j}^L \tag{13}$$

$$= K \sum_{i \in \mathcal{N}_j} \frac{n_i}{n} \frac{n_{i,j}}{n_i} \bar{c}_{i,j}^L \tag{14}$$

$$= K \sum_{i \in \mathcal{N}_j} \frac{1}{M} \frac{1}{K} \bar{c}_{i,j}^L \tag{15}$$

$$= \frac{1}{M} \sum_{i \in \mathcal{N}_j} \bar{c}_{i,j}^L \tag{16}$$

$$= \frac{1}{|\mathcal{N}_j|} \sum_{i \in \mathcal{N}_j} \bar{c}_{i,j}^L. \tag{17}$$

Eq. (15) holds when two conditions are met: (1) $\frac{n_i}{n} = \frac{1}{M}$, which occurs when all clients have the same number of samples, and (2) $\frac{n_{i,j}}{n_i} = \frac{1}{K}$, which is true when the local class distribution of each client is uniform. Eq. (17) follows from the condition that $|\mathcal{N}_j| = M$, meaning all clients have samples from all classes.

### B.2   Relationship between PPA and the Weighted-Averaging Method

Remark 1 also indicates that, under certain circumstances, the PPA method is equivalent to a scaled version of the weighted-averaging method described in Eq. (9), with a scaling factor of $\frac{1}{|\mathcal{N}_j|}$. We can establish this relationship through the following derivation:

$$\bar{c}_j^G = \frac{1}{|\mathcal{N}_j|} \sum_{i \in \mathcal{N}_j} \frac{n_{i,j}}{\sum_{i=1}^M n_{i,j}} \bar{c}_{i,j}^L, \tag{18}$$

$$= \frac{1}{|\mathcal{N}_j|} \sum_{i \in \mathcal{N}_j} \frac{\frac{n_i}{n} \frac{n_{i,j}}{n_i}}{\sum_{i=1}^M \frac{n_i}{n} \frac{n_{i,j}}{n_i}} \bar{c}_{i,j}^L \tag{19}$$

$$= \frac{1}{|\mathcal{N}_j|} \sum_{i \in \mathcal{N}_j} \frac{\frac{n_i}{n} \frac{n_{i,j}}{n_i}}{\sum_{i=1}^M \frac{1}{M} \frac{1}{K}} \bar{c}_{i,j}^L \tag{20}$$

$$= \frac{K}{|\mathcal{N}_j|} \sum_{i \in \mathcal{N}_j} \frac{n_i}{n} \frac{n_{i,j}}{n_i} \bar{c}_{i,j}^L \tag{21}$$

$$= \frac{1}{|\mathcal{N}_j|} \frac{K}{n} \sum_{i \in \mathcal{N}_j} n_{i,j} \bar{c}_{i,j}^L. \tag{22}$$

Eq. (20) holds when two conditions are met: (1) $\frac{n_i}{n} = \frac{1}{M}$, which occurs when all clients have the same number of samples, and (2) $\frac{n_{i,j}}{n_i} = \frac{1}{K}$, which is true when the local class distribution of each client is uniform.

## C   ALGORITHM OF FEDPROTO WITH CPS, PPA, AND CPKD

Our proposed components are designed for seamless integration into existing PBFL algorithms. To demonstrate this, we will outline the key modifications needed to incorporate these components into vanilla PBFL (FedProto). The first major change involves the server generating masking vectors and distributing them to clients, as shown in Line 1 and 8 of Algorithm 1. The second modification utilizes these vectors for exchanging compressed prototypes between the server and clients (Lines 5 and 11), followed by their reconstruction into structured sparse prototypes (Line 9). PPA and CPKD can be directly applied to FedProto by replacing its aggregation and distillation components. This integration process is similarly adaptable to other PBFL algorithms, such as FedTGP, showcasing the versatility of our methods across various PBFL frameworks.

When integrating PPA with FedTGP, careful attention to prototype scaling is crucial. FedTGP employs local prototypes to train a trainable prototype prior to aggregation. Applying PPA to FedTGP scales each local prototype by $n_{i,j}$, as shown in Eq. (11), which can lead to training loss divergence. To address this, we introduce a compensatory scaling factor. Specifically, we re-scale each local prototype by $\frac{K}{n} \cdot |\mathcal{N}_j|$, where $\mathcal{N}_j$ denotes the set of clients possessing samples from class $j$. This adjustment ensures stable training while preserving the benefits of both PPA and FedTGP.

---

**Algorithm 1** FedProto with CPS, PPA, and CPKD

---

**Input:** Number of client $M$, total communication rounds $T$, learning rate $\eta$, hyper-parameter $\lambda$
**Output:** Trained local models
 1: Initialize masking vector set $\{\boldsymbol{m}_j\}$ and compressed prototype set $\{\hat{c}_j^G\}$ for all classes.
 2: Initialize set $\mathcal{S}^0 = \{\}$ for clients selected up to the current iteration
 3: **for** iteration $t = 1, \ldots, T$ **do**
 4:     Server randomly samples a client subset $\mathcal{C}^t$
 5:     Server sends $\hat{c}_j^G$ to $\mathcal{C}^t$
 6:     **for** Client $i \in \mathcal{C}^t$ in parallel **do**
 7:         **if** $i \notin \mathcal{S}^{t-1}$ **then**
 8:             Server sends $\boldsymbol{m}_j$ to client $i$
 9:         Client $i$ reconstructs $\tilde{c}_j^G$ from $\hat{c}_j^G$ and updates its model with Eq. (5) and Eq. (12)
10:         Client $i$ computes $\bar{c}_{i,j}^L$ by Eq. (3) and convert it to $\hat{c}_{i,j}^L$
11:         Client $i$ sends $n_{i,j}\hat{c}_{i,j}^L$ to the server
12:     Server updates $\hat{c}_j^G$ with Eq. (11)
13:     Server updates $\mathcal{S}^t = \mathcal{S}^{t-1} \cup \mathcal{C}^t$
14: **return** Client models

---

## D   EXPERIMENTAL DETAILS

### D.1   HYPERPARAMETERS

For baseline algorithms, we adopt algorithm-specific hyperparameters as recommended in Zhang et al. (2024). Table 3 provides a comprehensive overview of these hyperparameter settings. It is important to note that the hyperparameter notations used in Table 3 are specific to each baseline method and may differ from notations used elsewhere in our paper.

Table 3: Hyperparameter settings for the compared methods.

| Method | Hyperparameter settings |
|---|---|
| LG-FedAvg | No additional hyperparameters |
| FML | $\alpha$ (KD weight for local model) $= 0.5$, $\beta$ (KD weight for meme model) $= 0.5$ |
| FedKD | $T_{\text{start}}$ (energy threshold) $= 0.95$, $T_{\text{end}}$ (energy threshold) $= 0.98$ |
| FedDistill | $\gamma$ (weight of logit regularizer) $= 1$ |

## D.2 CALCULATING COMMUNICATION COST

Table 4 presents formulations to calculate the communication cost per iteration shown in Table 1. $\boldsymbol{\theta}_{aux}$ and $\boldsymbol{\phi}_{aux}$ indicate the auxiliary feature extractor and classifier parameters, respectively.

Table 4: Formulation to calculate communication cost of algorithms.

| Algorithm | Communication cost | Algorithm | Communication cost |
|---|---|---|---|
| LG-FedAvg | $\sum_{i=1}^{M} \|\boldsymbol{\phi}_i\| \times 2$ | FedProto | $\sum_{i=1}^{M} d \times (K_i + K)$ |
| FML | $M \times (\|\boldsymbol{\theta}_{aux}\| + \|\boldsymbol{\phi}_{aux}\|) \times 2$ | FedProto+CPS | $\sum_{i=1}^{M} s \times (K_i + K)$ |
| FedKD | $M \times (\|\boldsymbol{\theta}_{aux}\| + \|\boldsymbol{\phi}_{aux}\|) \times 2 \times r$ | FedTGP | $\sum_{i=1}^{M} d \times (K_i + K)$ |
| FedDistill | $\sum_{i=1}^{M} K \times (K_i + K)$ | FedTGP+CPS | $\sum_{i=1}^{M} s \times (K_i + K)$ |

## D.3 EXPERIMENTAL ENVIRONMENT

To ensure reproducibility and provide a clear understanding of our experimental environment, we detail our setup below. Our experiments were designed to rigorously test the proposed methods under controlled conditions. The following list outlines the key components of our experimental setup:

- Framework: PyTorch 2.4
- Hardware:
  - CPUs: 2 Intel Xeon Gold 6240R (96 cores total)
  - Memory: 256GB
  - GPUs: Two NVIDIA RTX A6000
- Operating System: Ubuntu 22.04 LTS

This configuration allowed us to conduct our experiments efficiently and consistently, ensuring that our results are both reliable and reproducible. The code is provided in the supplementary materials.

# E VISUALIZATION OF PERSONALIZATION BY CPKD FOR CIFAR-100

We compute the Frobenius norm of the discrepancy between the data distribution heatmap and each FedProto heatmap. This computation results in a value of 6.44 for the standard FedProto implementation, while FedProto augmented with CPKD yields a lower value of 5.59.

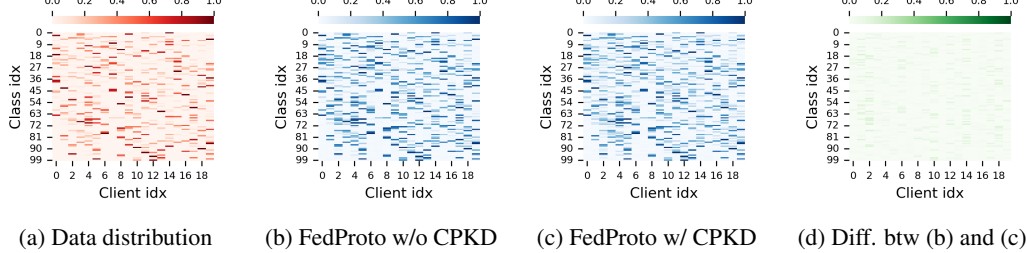

(a) Data distribution     (b) FedProto w/o CPKD     (c) FedProto w/ CPKD     (d) Diff. btw (b) and (c)

Figure 10: Heatmaps depicting the data distribution and L2-norms of class-specific weight vectors for the CIFAR-100 dataset.

