# OpenReview forum: "Enhancing Prototype-Based Federated Learning with Structured Sparse Prototypes"
_ICLR.cc/2025/Conference — ICLR 2025 Conference Withdrawn Submission_

### Official Review · Reviewer_y5jm · 2024-10-30

**Soundness:** 2
**Presentation:** 3
**Contribution:** 1
**Rating:** 3
**Confidence:** 4

**Summary:**

This paper proposes three new techniques to add onto Prototype-Based Federated Learning (PBFL) methods to (1) induce sparsity in order to reduce communication costs, (2) allow averaging amongst prototypes for each class without revealing information about each device's data distribution, namely the number of local data corresponding to a given class, and (3) personalize the knowledge distillation penalty by adding a weight corresponding to the number of data for a given prototype. Authors show that adding these methods to current PBFL reduces communication time and can marginally improve performance.

**Strengths:**

The authors tackle important issues within distributed learning: data and model heterogeneity. Furthermore, reducing the communication costs is crucial for edge settings where devices have limited memory.

The authors combine existing methods of sparsity/compression (CPS) and weighting (PPA & CPKD) to improve PBFL. The experimental results show improvement and promise (albeit some of it is marginal).

**Weaknesses:**

Unless I'm incorrect, the main improvements are the introduction of sparsity in the prototypes sent to the server and a new weighting term utilized within CPKD (Equation 12). These on their own are not enough of a contribution to warrant acceptance.

The method for private prototype aggregation (PPA) seems incorrect. There is no detail about how the normalization factor K is determined. I am unsure how a server could accurately determine this normalization factor without having $n_{i,j}$.

The new weighting term in CPKD is simply scaling by the proportion of samples from a given class. The exact term is $\beta_{i,j}=p_{i,j}/ \max_k p_{i,k}$. Why isn't $\beta_{i,j}$ simply equal to $p_{i,j}$? This seems more straightforward. Regardless, this seems like a very marginal adjustment to the underlying method.

**Questions:**

The introduction reads more like a related works section. The first paragraph briefly details some of the problems that the work is trying to solve before diving into what others do to solve the problem. Only by the third paragraph is there a description of what the authors are trying to explicitly solving. This should be bumped up earlier. The main challenges I believe are (1) server knowledge of device class distribution and (2) knowledge distillation of global prototype without considering data heterogeneity hinders personalization. This should be emphasized more.

I would discuss what prototypes are earlier in the paper. Even if very brief. I had to stop and read the original prototype paper (Tan et. al 2022) to understand what exactly are prototypes. I am used to FL but have not heard much of prototypes.

HPFL is mentioned but never defined. Is it simply just heterogeneous personalized FL?

Line 233: "where each element represents d/K consecutive dimensions of m for each class". Could the authors provide a more explicit explanation of how this is done? Is the element just the sum of the number of non-negative entries in the previous d/K entries?

As mentioned in the weaknesses, wow is the normalization factor K computed? Could the authors explain this?

Remark 1 is weak. It assumes an equal number of classes across all devices, an unrealistic setting. Of course in this scenario the server would know to choose K as $1/|N_j|$, but in any other setting what would K be?

When introducing CPKD, there is no justification. I found the discussion at the end of the Experiment Section. I would include this in Section 4 when detailed CPKD. I assume that the reason for the term is for more personalized prototypes. Less penalty on deviating prototypes when they don't play a major role in the device's dataset.

---

### Official Review · Reviewer_tumx · 2024-10-31

**Soundness:** 1
**Presentation:** 1
**Contribution:** 1
**Rating:** 3
**Confidence:** 4

**Summary:**

For prototype-based federated learning, the authors claim to address deficiencies in communication cost, privacy protection, and personalization by proposing three methods. Firstly, they apply structured sparsification to each class prototype to reduce communication costs. Secondly, to protect the client’s class distribution, they multiply the class prototypes by the class data on the client side before transmitting them to the server. Thirdly, they add class distribution weights during the distillation process to better match the client’s data distribution. Finally, the authors claim to have achieved significant advantages in their experiments.

**Strengths:**

When federated learning faces extreme heterogeneity in data distribution, the proposed Class-wise Prototype Sparsification in this paper can be used to reduce the communication cost between clients. However, this may come at the expense of sacrificing some performance.

**Weaknesses:**

1. The introduction and related work sections appear generated by LLM, for example, the first paragraph summarizes advances and limitations in federated learning concerning data and model heterogeneity, but the second paragraph abruptly shifts to interaction efficiency without elaborating on these challenges.

2. The claimed privacy and personalization benefits of PBFL lack empirical support, with vague statements like, "Moreover, PBFL enhances privacy protection by design because prototypes represent averages of local models’ representations. Furthermore, PBFL naturally facilitates personalization by allowing local models to distill knowledge exclusively from global prototypes corresponding to classes in their local datasets."

3. The paper reads more like a loosely assembled engineering report than a coherent scientific paper; the three methods proposed (CPS, PPA, CPKD) lack integration and even conflict with one another—CPKD requires direct access to P_ij the client’s class distribution, which contradicts PPA’s privacy goals. Additionally, I question the validity of PPA’s privacy claims, as there is no theoretical or empirical evidence supporting privacy or personalization benefits.

4. The impact of CPS sparsification on model performance lacks theoretical backing, and there’s no analysis to confirm if model convergence is preserved.

**Questions:**

1. The methods presented in this paper lack depth; they resemble engineering optimizations without the necessary theoretical basis or proofs. For example, while CPS demonstrates the feasibility of sparsification in prototype-based federated learning, there’s a need for further proof regarding its impact on performance and convergence.

2. The contribution to privacy and personalization has minimal impact and could be removed entirely. A focused exploration of sparsification’s contributions would be much stronger than the current scattered approach of three unrelated methods.

3. Reorganize the paragraph structure to clearly outline advances and challenges in federated learning with data and model heterogeneity, then gradually transition to issues of communication efficiency, privacy, and personalization.

4. The related work section offers only a superficial overview of heterogeneous federated learning. Research in this area covers a range of approaches, including regularization, bias suppression, distal terms, model splitting, decoupling, and weighted aggregation for personalization—yet the paper lacks references to this existing work.

---

### Official Review · Reviewer_XzVq · 2024-11-01

**Soundness:** 2
**Presentation:** 3
**Contribution:** 2
**Rating:** 3
**Confidence:** 5

**Summary:**

This submission proposes three approaches to address the existing problems in Prototype-Based Federated Learning (PBFL), including high communication costs for high-dimensional prototypes and numerous classes, privacy concerns during aggregation, and uniform knowledge distillation in heterogeneous data settings. Thanks a lot for the submission.  I did enjoy reading the paper as it tackles an important topic and is well-written.

**Strengths:**

To address the existing problems in Prototype-Based Federated Learning (PBFL), this paper proposes three methods;Classwise Prototype Sparsification (CPS), Privacy-Preserving Prototype Aggregation (PPA) and Class-Proportional Knowledge Distillation (CPKD). Overall, this article is easy to follow, but the contribution is limited.

--Important and interesting topic.
--Interesting and valuable insights (although, the most important ones have already been shown by related work).

**Weaknesses:**

This submission proposes three approaches to address the existing problems in Prototype-Based Federated Learning (PBFL), including high communication costs for high-dimensional prototypes and numerous classes, privacy concerns during aggregation, and uniform knowledge distillation in heterogeneous data settings.
Thanks a lot for the submission.  I did enjoy reading the paper as it tackles an important topic and is well-written. Unfortunately, most of its contribution has already been covered by prior work.

My main concern is that the main contribution of this paper, ‘Class-wise Prototype Sparsification’, is not distinct enough from existing prior work. Sparse learning strategies based on binary sparse masks have been widely studied, Li et al. (https://dl.acm.org/doi/10.1145/3485730.3485929) has already discussed this. In addition, Babakniya et al. (https://openreview.net/forum?id=iHyhdpsnyi) explain why sparse consensus is important in federated learning. Even if the given submission additionally introduces new methods that are distinct from those of Esipova et al.  I think that the most valuable insights are already published and new methods are not a large enough part of the paper to justify a conditional accept vote.
Another contribution of this submission is Class-Proportional Knowledge Distillation. To my knowledge, although correct and well motivated in this submission, this also has been shown by related work. Wu et al. (https://ieeexplore.ieee.org/abstract/document/10163770) and Dai et al. (https://ojs.aaai.org/index.php/AAAI/article/view/25891) have presented a similar or more advanced approach to this paper.
In summary, I think that the insights already gathered by prior work cover the largest part of this submission.  Unfortunately, this means that for acceptance, a major part of the submission would have to be redesigned which is too much to be fixed in a major revision.


Apart from the issues related to prior work, I had some other issues that could be fixed in the future:
The submission lacks a summary overview diagram, adding a clear， readable overview diagram would be more conducive to readers’ understanding of the workflow and detail of these proposed enhanced PBFL approaches.
I suggest that the author conduct comparative experiments on devices with different bandwidths to demonstrate the practicality of these enhanced PBFL approaches may more practical in resource-constrained environments.
FedTGP combined with Privacy-preserving Prototype Aggregation alone, which shows the highest classification accuracy. Why does PPA, designed for enhanced data protection, improve the performance of federated learning?

Add discussion of and distinction from related work, especially Li et al. (https://dl.acm.org/doi/10.1145/3485730.3485929). Show if contribution is still valuable.
Add a summary overview diagram.
Additional experiments to prove the practicality of these enhanced PBFL approaches may more practical in resource-constrained environments.

Contribution not distinct enough from existing prior work, sparse learning strategies based on binary sparse masks have been discussed in related work by Li et al. (https://dl.acm.org/doi/10.1145/3485730.3485929).
Reasoning why sparse consensus is important in federated learning has already been shown by related work by Babakniya et al. (https://openreview.net/forum?id=iHyhdpsnyi), does provide no new insights.

**Questions:**

This submission proposes three approaches to address the existing problems in Prototype-Based Federated Learning (PBFL), including high communication costs for high-dimensional prototypes and numerous classes, privacy concerns during aggregation, and uniform knowledge distillation in heterogeneous data settings.
Thanks a lot for the submission.  I did enjoy reading the paper as it tackles an important topic and is well-written. Unfortunately, most of its contribution has already been covered by prior work.

My main concern is that the main contribution of this paper, ‘Class-wise Prototype Sparsification’, is not distinct enough from existing prior work. Sparse learning strategies based on binary sparse masks have been widely studied, Li et al. (https://dl.acm.org/doi/10.1145/3485730.3485929) has already discussed this. In addition, Babakniya et al. (https://openreview.net/forum?id=iHyhdpsnyi) explain why sparse consensus is important in federated learning. Even if the given submission additionally introduces new methods that are distinct from those of Esipova et al.  I think that the most valuable insights are already published and new methods are not a large enough part of the paper to justify a conditional accept vote.
Another contribution of this submission is Class-Proportional Knowledge Distillation. To my knowledge, although correct and well motivated in this submission, this also has been shown by related work. Wu et al. (https://ieeexplore.ieee.org/abstract/document/10163770) and Dai et al. (https://ojs.aaai.org/index.php/AAAI/article/view/25891) have presented a similar or more advanced approach to this paper.
In summary, I think that the insights already gathered by prior work cover the largest part of this submission.  Unfortunately, this means that for acceptance, a major part of the submission would have to be redesigned which is too much to be fixed in a major revision.


Apart from the issues related to prior work, I had some other issues that could be fixed in the future:
The submission lacks a summary overview diagram, adding a clear， readable overview diagram would be more conducive to readers’ understanding of the workflow and detail of these proposed enhanced PBFL approaches.
I suggest that the author conduct comparative experiments on devices with different bandwidths to demonstrate the practicality of these enhanced PBFL approaches may more practical in resource-constrained environments.
FedTGP combined with Privacy-preserving Prototype Aggregation alone, which shows the highest classification accuracy. Why does PPA, designed for enhanced data protection, improve the performance of federated learning?

Add discussion of and distinction from related work, especially Li et al. (https://dl.acm.org/doi/10.1145/3485730.3485929). Show if contribution is still valuable.
Add a summary overview diagram.
Additional experiments to prove the practicality of these enhanced PBFL approaches may more practical in resource-constrained environments.

Contribution not distinct enough from existing prior work, sparse learning strategies based on binary sparse masks have been discussed in related work by Li et al. (https://dl.acm.org/doi/10.1145/3485730.3485929).
Reasoning why sparse consensus is important in federated learning has already been shown by related work by Babakniya et al. (https://openreview.net/forum?id=iHyhdpsnyi), does provide no new insights.

---

### Official Review · Reviewer_h15y · 2024-11-04

**Soundness:** 3
**Presentation:** 3
**Contribution:** 2
**Rating:** 5
**Confidence:** 4

**Summary:**

This manuscript proposed a prototype-based personalized federated learning approach, which primarily aims at reducing communication cost in a resource constraint environment.  This is achieved by reducing dimensions of the protypes during their sharing with the server. In addition to this, the authors have also addressed privacy  and heterogeneity issues. In a nutshell, a class-wise prototype sparsification method is used to reduce the communication cost, a new aggregation method  is introduced to enhance privacy, and finally, a class-proportional knowledge distillation method is introduced to improve personalization.

**Strengths:**

1. This paper introduced three key methods to enhance performance of PBFL, while reducing communication cost, enhancing privacy, and improving personalization.
2. The methods are easy to integrate with existing PBFL approaches.
3. The results showed that the performance of the proposed methods are comparable with the baselines.

**Weaknesses:**

1. The proposed CPS method seems very straightforward. There exist several well-known sparsity reduction techniques (see below references) in the literature. it would be nice if the author can compare the proposed CPS method with the existing ones.

 Wen, W., Wu, C., Wang, Y., Chen, Y., & Li, H. (2016). Learning structured sparsity in deep neural networks. Advances in neural information processing systems, 29.

Han, S., Mao, H., & Dally, W. J. (2015). Deep compression: Compressing deep neural networks with pruning, trained quantization and huffman coding. arXiv preprint arXiv:1510.00149.)

2. There is no proof of convergence or any theoretical supports. In particular, a formal privacy guarantee for the PPA method and a mathematical justification for the effectiveness of CPKD would be helpful.
3. The experiments are quite limited, lacking tests in various heterogeneous and different federated learning settings. For example, the authors may consider reducing the feature dimensions  to 64, 128, or 256, and test with different levels of data heterogeneity (i.e., α=0.05,0.5,0.1)
4. There is no comparison with non-PBFL approaches that aim for similar objectives.
5. There is little discussion regarding the computational overhead introduced by the new methods. The authors may provide the computation complexity of their methods either empirically or theoretically.

**Questions:**

1. In Section 4.1, you mention only about ReLU activation function which can lead to dead hidden units. What are the effects of other activation functions, as some classification model architectures may use activation functions other than ReLU?
2. To address privacy concerns, you have modified the prototype aggregation function and you mentioned that there is no need of client to share n_i. I am curious to know how do you compute n in Equation 11 without knowing total sample of individual client i?
3. What is the mathematical rationale behind the effectiveness of CPKD?
4. What is the computational overhead associated with implementing all three methods?
5. In your experiments, you have set the feature dimension to 500 and observe numerous dead points. What would be the impact of reducing the feature dimensions (e.g., 64, 128, or 256)? It would be interesting to see the findings experimentally.
6. Your experiments assume full participation from clients. How would partial client participation affect the performance of your proposed methods?
7. In your experimental setup, you mentioned the use of different model architectures, but I cannot find those results. It would be interesting to see the findings illustrating the effectiveness of your proposed method across varying architectures?
8. What would happen if the methods are tested with different levels of data heterogeneity (e.g., α=0.05,0.5,0.1)?
9. For a more comprehensive understanding of your method's superiority, I would suggest to include a standard non-PBFL baseline such as SCAFFOLD.

---

### Note · Authors · 2024-11-18

**Comment:**

Thank you for your thorough review of our paper and for providing such thoughtful and detailed feedback. We fully agree with the reviewers' suggestions, which have motivated us to further improve our research. We will continue our work with your valuable comments in mind. We sincerely appreciate the time you took to review our paper.

**Withdrawal Confirmation:**

I have read and agree with the venue's withdrawal policy on behalf of myself and my co-authors.